# Comparison of Different Linewidth Measuring Methods for Narrow Linewidth Laser

**DOI:** 10.3390/s23010122

**Published:** 2022-12-23

**Authors:** Ziqi Zheng, Qiaoxia Luo, Xian Wang, Xiaohui Ma, Wei Zhang, Wentan Fang, Xiaolin Chen, Song Huang, Yong Zhou, Weiqing Gao

**Affiliations:** Department of Optical Engineering, School of Physics, Hefei University of Technology, Feicui Road 420, Hefei 230601, China

**Keywords:** single longitudinal mode laser, self-injection locking, stimulated Brillouin scattering, delayed self-heterodyne interferometry, self-correlation envelope, Voigt fitting

## Abstract

We experimentally demonstrate a fiber laser with different linewidths based on self-injection locking (SIL) and the stimulated Brillouin scattering effect. Based on the homemade fiber laser, the error origin, resolution, and applicable range of delayed self-heterodyne interferometry (DSHI), self-correlation envelope linewidth detection (SCELD) and Voigt fitting are investigated numerically and experimentally. The selection of the linewidth measuring method should meet the following conclusions: an approximately Lorentzian self-heterodyne spectrum without the pedestal and high-intensity sinusoidal jitter is a prerequisite for DSHI; the SCELD needs a suitable length of delay fiber for eliminating flicker noise and dark noise of the electrical spectrum analyzer; a non-Lorentzian self-heterodyne spectrum without a pedestal is an indispensable element for Voigt fitting. According to the experimental results, the laser Lorentzian linewidth of SIL changes from 1.7 kHz to 587 Hz under different injection powers. When the Brillouin erbium fiber laser is utilized, the Lorentzian linewidth is measured to be 60 ± 5 Hz.

## 1. Introduction

Recently, single longitudinal mode (SLM) lasers have attracted increasing attention due to excellent characteristics, such as narrow spectral linewidth, long coherent length, and low intensity noise, which make them irreplaceable in the fields of coherent beam combining, high-rate optical coherent communications, and fiber sensing [1,2,3]. Although SLM lasers possess multiple merits, the linewidth compression to obtain a lower frequency noise has still aroused great interest. The lasing with an ultra-narrow linewidth has great potential for some advanced fields, such as gravitational wave detection and the high-precision optical measurement system [4,5,6].

According to previous research, lasing with an ultra-narrow linewidth can be achieved by many methods, such as slow light effect [7] and frequency stabilization technology based on Pound–Drever–Hall (PDH) or optical frequency comb [8,9,10]. However, they are difficult to widely apply because of the complicated structures and high cost. Besides, there are two efficient and practical methods. One uses self-injection locking (SIL) and the other is based on the stimulated Brillouin scattering (SBS) effect. The SIL offers a simple and effective approach to compressing the laser linewidth by the high-Q-value external resonator [11,12]. The feedback effect caused by the SIL can be divided into five regimes according to the injected light power launched into the main cavity [13]. A fascinating fact is that the laser linewidth is sensitive to the injection power in the third regime of the feedback effect. Therefore, the laser linewidth can be adjusted by the injection power [13]. The SBS effect can realize the SLM operation due to the ultra-narrow gain range and generate an ultra-narrow laser linewidth because of the high Q value [14]. Normally, a long single-mode fiber (hundreds of meters) is necessary for exciting SBS because of the small Brillouin gain coefficient. However, mode hopping inevitably occurs in a Brillouin laser with such a long cavity. Fortunately, the Brillouin erbium fiber laser (BEFL) has solved this problem well [15,16].

With the development of laser linewidth narrowing technology, researchers began to pay close attention to the measuring methods of laser linewidth. The heterodyne beat method can provide high measuring precision; however, this method requires an extra lasing with ultra-narrow linewidth and close frequency to the lasing under test, which is hard to be achieved. The *β*-separation line method can measure the laser linewidth from the frequency noise spectral density, but the measurement of frequency noise spectral density is complicated [17,18]. Compared with the methods mentioned above, the delayed self-heterodyne interferometry (DSHI), the self-correlation envelope linewidth detection (SCELD), and the Voigt fitting show a simpler operation and wider measuring range. Since first demonstrated by Okoshi et al. [19], the DSHI technique has been an important tool for laser linewidth measuring. The analysis of error origin still appears incomplete although the DSHI has been widely studied. Huang et al. introduced the detailed principle and experimental process of SCELD in Ref. [20] but did not provide a discussion on the resolution. Chen et al. innovatively proposed the Voigt fitting for ultra-narrow linewidth measuring, but also lacked a discussion on the applicable scope [21]. In addition, the previous researchers prefer the precision of the measuring technology they used rather than proposing the relevance between the three measuring technologies.

In this work, we experimentally realized an SLM fiber laser with different linewidths based on the SIL and BEFL. The error origin, resolution, and applicable range of the DSHI, SCELD, and Voigt fitting were investigated numerically and experimentally based on the homemade fiber laser. From the elaboration of the three methods, the selection of the linewidth measuring method should meet the following principles: an approximately Lorentzian self-heterodyne spectrum without the central pedestal and sinusoidal jitter is a prerequisite for the DSHI method; proper delay fiber length for the SCELD method is important to avoid the influence of the dark noise of the electrical spectrum analyzer (ESA) and flicker noise at the same time; for the Voigt fitting method, the self-heterodyne spectrum without the central pedestal is an indispensable element for precise narrow linewidth measurement. Finally, according to the experimental results, the laser Lorentzian linewidth of the SIL laser under different injection powers was measured by the DSHI and SCELD to be from 1.7 kHz to 587 Hz. The Lorentzian linewidth of BEFL was confirmed to be 60 ± 5 Hz measured by the Voigt fitting.

## 2. Experimental Setup

The experimental setup of the distributed feedback (DFB) laser with the SIL is shown in Figure 1a. The 980 nm pump was launched into the DFB through a wavelength-division multiplexer (WDM1). The backward signal was injected into the following amplifier through an isolator (ISO), an optical coupler (OC1, 99:1), and WDM3. The residual pump from the DFB was drawn from WDM2 and used for the following amplifier. The external resonator was composed of a variable optical attenuator (VOA1) for the injection power adjustment, a single-mode fiber (SMF) with a length of 20 m for high Q value, and an OC2 with a coupling ratio of 1% for the injection power detection. The amplified spontaneous emission from the amplifier was removed by the fiber Bragg grating (FBG), and the reflection spectrum of the FBG is shown in Figure 1d. The central wavelength and 3-dB spectral bandwidth are 1550.1 nm and 0.19 nm, respectively. Figure 1b illustrates the experimental setup of the BEFL. The output lasing from Figure 1a provided the Brillouin pump (BP) for the BEFL through a circulator (CIR2). The OC3 with a coupling ratio of 5:5 was used for the measurement of BP. The VOA2 was used to adjust the BP power. A 1-m-long erbium-doped fiber (EDF2, LIEKKI Er-80) was pumped by the second 980 nm LD through the WDM4 to act as the gain media for the BEFL. The tunable optical filter (TOF, Santec OTF-320) with a 3-dB bandwidth of 0.8 nm was used to suppress the self-excited oscillation of BEFL. The lasing was emitted from the OC4 with a coupling ratio of 30%.

## 3. Results

### 3.1. Spectral Properties

During the experiment, the 980 nm pump power of the SIL laser was fixed at the maximum of 390 mW. The self-injection power was controlled by VOA1 and limited to 1.4 μW (measured from the 1% port of the OC2) since the stronger injection power would lead to mode hopping in the main cavity. The side-mode suppression ratio (SMSR) of the SIL laser was measured by the DSHI, as shown in Figure 1c. The SMSR under maximum injection power (1.4 μW) was measured to be 71 dB, as shown in Figure 2b. The laser spectrum of the SIL laser was measured by an optical spectrum analyzer (OSA, Yokokawa AQ6375) with a maximum resolution of 0.05 nm, which shows the central wavelength of 1550.05 nm, as shown in Figure 2a. A scanning Fabry–Perot interferometer (FPI, Thorlabs SA200-12B, resolution of 67 MHz) was used to measure the longitudinal-mode spectrum. The single-frequency operation was confirmed according to the inset of Figure 2a The BEFL emitted the lasing with the maximum power when the 980 nm pump power of the BEFL laser exceeded 210 mW. Figure 2c illustrates the spectrum of the BEFL. The central wavelength interval is 0.05 nm compared to the BP (limited by the OSA resolution). We injected the BP and BEFL lasing into the FPI at the same time to confirm the frequency interval between them. The frequency interval was measured to be 10.82 GHz, which corresponds to the Brillouin frequency shift at 1.5 μm, as shown in Figure 2d.

### 3.2. Linewidth Measurement with DSHI

Flicker noise and white noise are two major factors affecting the self-heterodyne spectrum. The white noise is caused by spontaneous emission, which is independent of delay time. However, the flicker noise is positively correlated with the delay time. Since most applications are not sensitive to the flicker noise, researchers mainly focus on the laser white noise linewidth. When only the white noise is considered, the self-heterodyne spectrum can be expressed as [22]
(1)$$S_{\left(\omega \right)}=S_{1}+S_{2}•S_{3}$$
(2)$$S_{1}={\left(1+\alpha^{2}\right)}^{2}\delta \left(\omega \right)+2\alpha^{2}e^{-S_{0}\tau_{0}}\delta \left(\omega -\Omega \right)$$
(3)$$S_{2}=\frac{4\alpha^{2}S_{0}}{{\left(S_{0}{}^{2}+{\left(\omega -\Omega \right)}^{2}\right)}^{}}$$
(4)$$S_{3}=1-e^{-S_{0}\tau_{0}}\left(\cos\left(\left(\omega -\Omega \right)•\tau_{0}\right)+\frac{S_{0}}{\omega -\Omega }•\sin\left(\left(\omega -\Omega \right)•\tau_{0}\right)\right)$$
(5)$$S_{0}=\frac{f}{\pi }=\frac{1}{2\pi^{2}\tau }$$
where *w* is the angular frequency, *α* is the amplitude ratio between the two branches, *τ*_0_ is the delay time, *τ* is the laser coherent time, *Ω* is the frequency-shift of the AOM, δ denotes the Dirac delta function, $S_{0}$ is the single-sided spectral density of the white noise, and *f* is the linewidth caused by white noise.

Figure 3a–d illustrates the simulated self-heterodyne spectrum of the DSHI, when the delay time is 0.006-, 0.06-, 0.6-, and 6-times laser coherent time, respectively. When the delay time is short, the main energy of the self-heterodyne spectrum will focus on the central pedestal as described in Equation (2). The central pedestal will narrow with the delay time increase and shift the energy to the modified Lorentzian part as described in Equation (3), as shown in Figure 3a–c. The self-heterodyne spectrum will evolve into a standard Lorentzian lineshape under sufficient delay time, as shown in Figure 3d. According to Equation (4), the intensity of the sine function will decrease with the delay time increasing, as shown in Figure 3a–c. The pedestal and the sinusoidal jitter can be regarded as caused by the measurement error since they cannot be eliminated unless the delay time is greater than six times the laser coherent time [23], as shown in Figure 3d. Figure 3e,f shows the relative measurement error with different delay time. The laser linewidth was calculated from the 3-dB and 20-dB bandwidth of the self-heterodyne spectrum. The peaks in Figure 3e are related to the sinusoidal jitter, as shown in the inset. According to Figure 3e,f, the relative measurement error can be ignored when the white noise linewidth is calculated by the 20-dB bandwidth and the delay time is close to the laser coherent time. However, the coherent time of a narrow-linewidth lasing is quite long, which means the flicker noise will become prominent when the delay time approaches such a long coherent time. At this time, the self-heterodyne spectrum can be recognized as the convolution of the flicker noise (wide Gaussian lineshape) and white noise (narrow Lorentzian lineshape), which makes it difficult to obtain the white noise linewidth (Lorentzian linewidth).

The self-heterodyne spectra of the SIL laser with different injection powers were observed by the DSHI (50 km delay fiber), as shown in Figure 4a. The laser linewidth was narrowed with the injection power increasing, which suggested that the injection power is in the third regime of the feedback effects [13]. When the self-injection was removed, the self-heterodyne spectrum showed a 20-dB bandwidth of 34 kHz, indicating the laser linewidth of 1.7 kHz (coherent length: 18.7 km silica fiber), as shown in Figure 4b. Two Lorentzian curves calculated from the 3-dB and 20-dB bandwidth of the self-heterodyne spectrum are shown in Figure 4b. The Lorentzian curve of the 20-dB bandwidth agrees well with the experimental results since the flicker noise effect is mainly concentrated in the spectrum center. Figure 4c illustrates the self-heterodyne spectrum with maximal injection power of 1.4 μW and the two Lorentzian curves. The 20-dB bandwidth is 15 kHz corresponding to the laser linewidth of 750 Hz (coherent length: 42.5 km silica fiber). It can be confirmed that the flicker noise effect begins to appear since the 20-dB Lorentzian curve is wider than the measured spectrum. Hence, the real Lorentzian linewidth is less than 750 Hz. The self-heterodyne spectrum of BEFL and the two Lorentzian curves are shown in Figure 4d. Obviously, neither of the two Lorentzian curves can reproduce the self-heterodyne spectrum, which indicates that DSHI cannot estimate the Lorentzian linewidth of the BEFL since the flicker noise completely covers the white noise. The results imply that a more advanced method is necessary for accurate measurement of a narrow linewidth laser.

### 3.3. Linewidth Measurement with SCELD

The experimental setup of the SCELD is the same as the DSHI in Figure 1c but with a shorter delay fiber. The Lorentzian linewidth was calculated by the contrast difference between the second peak and the second trough (CDSPST) of the self-heterodyne spectrum. It can be expressed as [20]
(6)$$\Delta S=10\log_{10}\frac{\left(1+{\left(\frac{2c}{nfL}\right)}^{2}\right)\left(1+e^{\frac{{}^{-2\pi nfL}}{c}}\right)}{\left(1+{\left(\frac{3c}{2nfL}\right)}^{2}\right)\left(1-e^{\frac{-2\pi nfL}{c}}\right)}$$
where Δ*S* is the CDSPST, *c* is the light speed, *n* is the refractive index, *f* is the Lorentzian linewidth of the under-test lasing, and *L* is the delay fiber length. From Equation (6), the Δ*S* changes slightly when the wide-linewidth lasing is under test. The variation will be even lower than the intensity resolution of the ESA. Figure 5 illustrates the CDSPST and variation rate curves versus the under-test laser linewidth. It can be recognized that the CDSPST has a negative correlation with the laser linewidth. The variation rate decreases rapidly with the laser linewidth increasing, which means that only the narrow laser linewidth can obtain high measurement precision.

A long delay fiber will cause a larger Lorentzian linewidth value than reality because the prominent flicker noise improves the trough intensity and results in a smaller CDSPST, as shown in Figure 6a. However, a short delay fiber will also lead to a measurement error due to the weak trough intensity which may touch the dark noise of the ESA. Hence, the delay fiber with appropriate length is key to accurate linewidth measurement in the SCELD. The self-heterodyne spectrum of the SIL laser under maximal injection power was measured by the SCELD with a 4.168 km delay fiber, as shown in Figure 6b. The Δ*S* is 16.65 dB corresponding to the product *fL* of 2.449 MHz∙m and the Lorentzian linewidth of 587 Hz, which is slightly smaller than 750 Hz in Figure 4c as expected. The simulated self-heterodyne spectrum shows high consistency with the experimental results except for two mismatch regions, which ensures the measurement precision of the linewidth. The first mismatch region near 80 MHz (marked by the pink arrow in Figure 6b) is due to the omission of *S*_1_ during simulation, and the second mismatch region beyond 80.15 MHz (marked by the green arrow in Figure 6b) is due to the black noise of the ESA. The second mismatch also indicates that the delay fiber length cannot be reduced for the laser under test due to the dark noise of the ESA. The self-heterodyne spectrum of BEFL was measured by the SCELD with a 14.3 km delay fiber, as shown in Figure 6c. The Δ*S* is 17.04 dB corresponding to the product *fL* of 2.233 MHz∙m and the Lorentzian linewidth of 156 Hz. The simulated self-heterodyne spectrum shows a large mismatch with the experimental results, which means a large measurement error. The first mismatch region near 80 MHz (marked by the pink arrow in Figure 6c) and the second mismatch region beyond 80.3 MHz (marked by the green arrow in Figure 6c) are caused by the same reason in Figure 6b. The mismatch (about 5 dB) at the first trough (marked by the blue arrow in Figure 6c) is due to the flicker noise which cannot be filtered out with such a long delay fiber of 14.3 km. From Figure 6c, neither a longer nor shorter delay fiber is available for the linewidth measurement of BEFL with SCELD due to the flicker noise and dark noise.

### 3.4. Linewidth Measurement with Voigt Fitting

For the lasing with the linewidth near or even narrower than 100 Hz, the white noise is much weaker than the flicker noise, which means the Lorentzian linewidth cannot be obtained directly. The measured self-heterodyne spectrum is the Voigt spectral profile, which is the convolution of the white noise with the Lorentzian lineshape and the flicker noise with the approximately Gaussian lineshape. The accurate estimate of the Lorentzian linewidth can be obtained by fitting the Voigt curve to the measured self-heterodyne spectrum [21,22].

An excessively long fiber is not suitable for obtaining a self-heterodyne spectrum since it will lead to a large power loss and strong nonlinearity. However, a short delay fiber may cause an additional error, such as a prominent pedestal and strong sinusoidal jitter. In order to find out a proper delay fiber length for the Lorentzian linewidth measurement of the BEFL, the self-heterodyne spectra of the BEFL with different delay fiber lengths were measured, as shown in Figure 7. The results illustrate that the central pedestal narrows with the delay fiber increasing, and finally vanishes when the delay fiber is 20 km. The vanishment of the central pedestal implies that the self-heterodyne spectrum can be accurately fitted with the Voigt curve.

The fitting process is as flows: the Lorentzian linewidth $\upsilon_{L}$ is firstly evaluated from the 20-dB bandwidth of the self-heterodyne spectrum; the Gaussian linewidth $\upsilon_{G}$ is calculated from Equation (7) with the measured Voigt 3-dB bandwidth $\upsilon_{V}$; the fitted Voigt curve is the convolution of the Lorentzian and Gaussian curves; $\upsilon_{L}$ will be refined from the fitted Voigt curve. The process will be iterated until the fitted Voigt curve bandwidth is close to the 20 dB bandwidth of the measured Voigt spectrum (the error range was set to 5 Hz during the simulation).
(7)$$\upsilon_{V}=\frac{1}{2}\left(1.0692\upsilon_{L}{}^{2}+\sqrt{0.866639\upsilon_{L}{}^{2}+4\upsilon_{G}{}^{2}}\right)$$

The self-heterodyne spectrum of the BEFL measured by the DSHI (20 and 50 km) and the fitted curves are presented in Figure 8a,b. The Voigt fitting curves reproduce the self-heterodyne spectrum well and the Lorentzian linewidth is confirmed to be 60 ± 5 Hz. The results show that a self-heterodyne spectrum without the central pedestal is an indispensable element for an accurate narrow linewidth measurement.

## 4. Discussion

The DSHI shows clear advantages of the simple measuring device and fast data processing. However, the measuring range of the DSHI must be limited above kHz, because a short delay fiber will lead to a heterodyne spectrum with obvious sinusoidal jitter and a long delay fiber will enhance the flicker noise. The main purpose of the SCELD is to eliminate the flicker noise with a short delay fiber. The linewidth can be retrieved as described in Equation (6) to avoid the effect of the central pedestal and sinusoidal jitter. The SCELD can measure any Lorentzian linewidth under ideal conditions. However, the accuracy depends on the intensity resolution and dark noise of the ESA as shown in Figure 6, which makes it difficult to measure the wide linewidth lasing with SCELD (less than hundreds of Hz with common ESA). A suitable delay fiber length needs to be estimated with a probable linewidth and the performance of the ESA. For Voigt fitting, the effect of the flicker noise can be avoided by separating the two kinds of noises as described in Section 3.4. This enables it to measure the ultra-narrow linewidth with high accuracy and low device cost. Whereas, due to the high demand for the Voigt spectral profile of the self-heterodyne spectrum, the delay fiber length needs to be confirmed within a suitable range. The Lorentzian linewidth is not obtained directly from the self-heterodyne spectrum and needs to be fitted out through compacted iterations.

## 5. Conclusions

In conclusion, a fiber laser with tunable linewidth based on the SIL and SBS effect has been presented. The linewidth-tuning operation was achieved by the injection power adjustment within the third regime of the feedback effect. The BEFL with an ultra-narrow linewidth was generated by the narrow gain range of the SBS effect. Meanwhile, three kinds of linewidth measuring methods were analyzed numerically and experimentally based on the homemade fiber laser. The principle of linewidth measuring method selection is summarized as: (1) For the DSHI method, the application range should be above kHz and the self-heterodyne spectrum should meet an approximately Lorentzian shape without the central pedestal and sinusoidal jitter under a long enough delay fiber; (2) For the SCELD method, a proper delay fiber length is important to realize a high compliance between the measured and retrieved self-heterodyne spectra, the application range should be above hundreds Hz with a common ESA; (3) For the Voigt fitting method, the delay fiber should be long enough to avoid the central pedestal on the self-heterodyne spectrum, the application range should be above dozens Hz. The results can be helpful to understand the noise principle better in optical fiber sensors and be used to promote sensing accuracy.

## Figures and Tables

**Figure 1 sensors-23-00122-f001:**
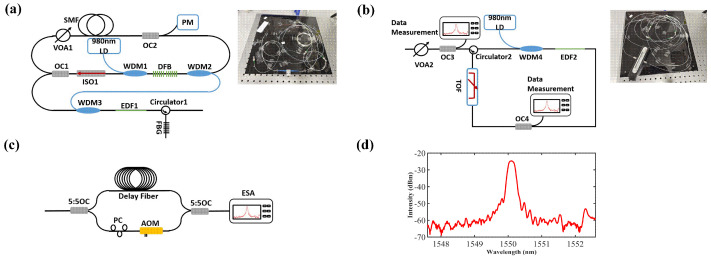
Experimental setup of (**a**) DFB with SIL, (**b**) BEFL, and (**c**) DSHI. (**d**) Reflection spectrum of the FBG. PM: power meter; PC: polarization controller; AOM: acoustic optic modulator. Insets: the photographs of the experiment setup.

**Figure 2 sensors-23-00122-f002:**
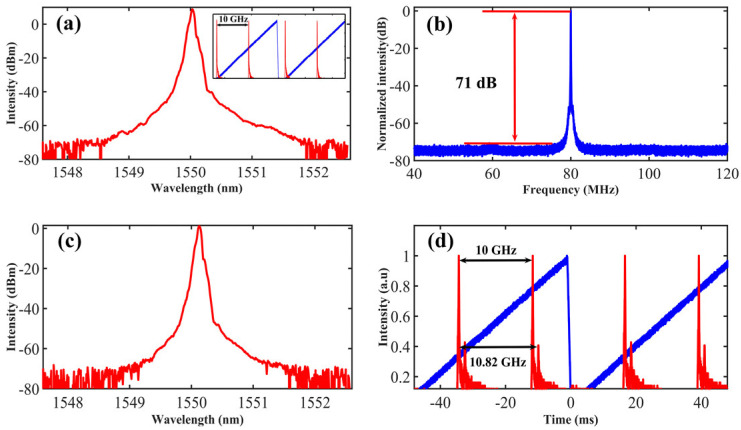
Spectrum of (**a**) SIL laser (Inset: FPI spectrum) and (**c**) BEFL. (**b**) Frequency domain properties of the SIL laser under maximum injected power. (**d**) FPI spectrum integrated with the BP and BEFL lasing, blue line: normalized voltage loaded to FPI.

**Figure 3 sensors-23-00122-f003:**
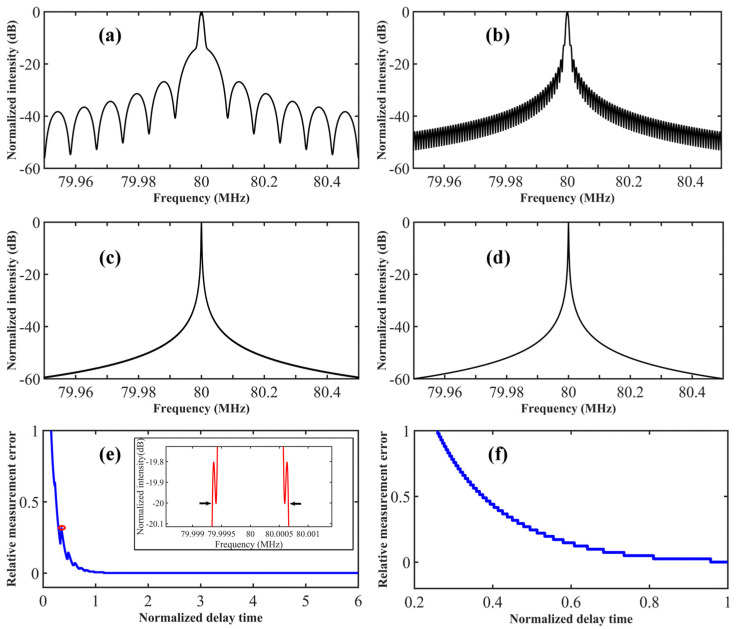
The simulated self-heterodyne spectrum when the ratio of delay time to coherent time is (**a**) 0.006, (**b**) 0.06, (**c**) 0.6, and (**d**) 6. The relative measurement error with different delay time when the laser linewidth is calculated from the (**e**) 20-dB bandwidth and (**f**) 3-dB bandwidth. Inset, the zoomed in self-heterodyne spectrum corresponding to the read mark. Parameters used in simulation: *f* = 50 Hz, *n* = 1.5, *Ω* = 80 MHz.

**Figure 4 sensors-23-00122-f004:**
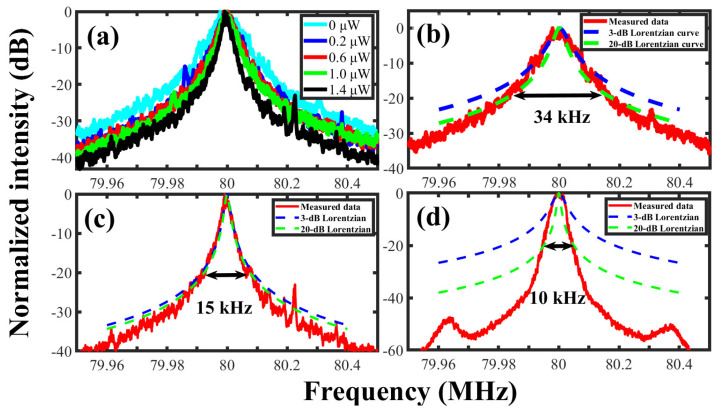
The self-heterodyne spectrum and Lorentzian curves. (**a**) The SIL laser under different injection powers, (**b**) the SIL laser without injection power, (**c**) the SIL laser with maximal injection power, (**d**) the BEFL.

**Figure 5 sensors-23-00122-f005:**
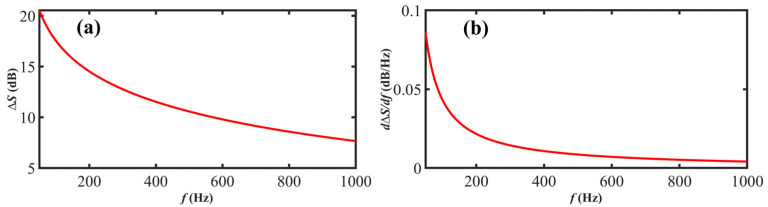
(**a**) The CDSPST and (**b**) variation rate versus under-test laser linewidth. Parameters used in simulation: *L* = 20 km, *n* = 1.5.

**Figure 6 sensors-23-00122-f006:**
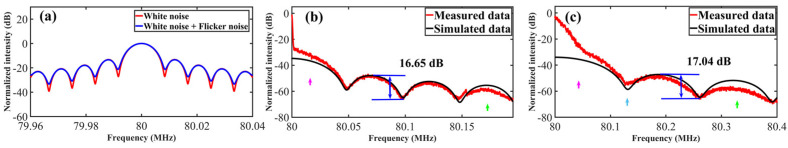
(**a**) The simulated self-heterodyne spectrum with and without the flicker noise. The self-heterodyne spectrum and simulated curve of (**b**) the SIL laser with delay fiber of 4.168 km and (**c**) the BEFL with delay fiber of 14.3 km.

**Figure 7 sensors-23-00122-f007:**
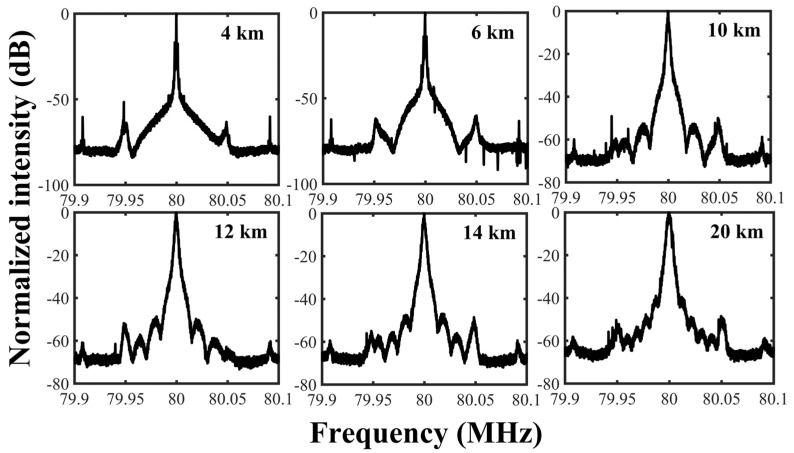
The self-heterodyne spectrum of BEFL with different delay fiber length.

**Figure 8 sensors-23-00122-f008:**
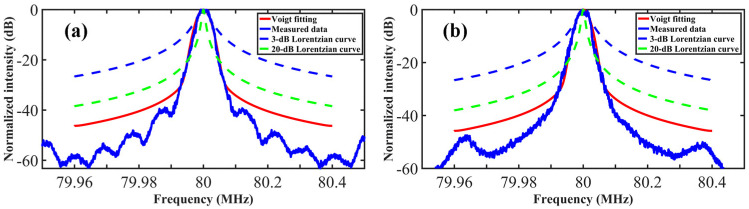
The self-heterodyne spectrum of BEFL and Voigt fitting curves. (**a**) 20 km and (**b**) 50 km delay fiber.

## Data Availability

Data will be made available on request.

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
