# Peer review of "Comparison of Different Linewidth Measuring Methods for Narrow Linewidth Laser"

_sensors, 2022, doi:10.3390/s23010122_

Round 1
Reviewer 1 Report
Very nice paper, it is well written. I've two minor comments,
1. No dots after Figure/Figures
2. Line 170: can't --> can not
Reviewer 2 Report
The article is about Research on the Different Linewidth Measuring Methods for Narrow Linewidth Laser.
There are only a few remarks/questions to be clarified by the authors, namely:
1. There exist some undefined acronyms in the text, which need to be properly declared.
2. This paper's abstract needs to be revised in light of current difficulties and the new methodology's solution.
3. Please explain the simulation and experimental platforms in more detail so the reader can easily understand them. Please replace Figure 1 with a photograph of the experiment. it would add more value to this article.
4. Line 95-96: Please explain why these limitations are at 310 mW and 1.4 µW?
5. What are the directions for further research (e.g. measurements on real objects under normal environmental conditions).?
Reviewer 3 Report
In this work, authors demonstrated a SIL to induce SBS effect. Successively, 3 linewidth measurement methods are used for further analysis. Scientifically, I don't find any significant mistakes. Some comments to improve the manuscript as below:
1. The title is too general! "Research on...." not really appropriate for this manuscript. An example, "Comparison of linewidth measurement method...." may be more appropriate.
2. Figure 1 is too small to read, please improve the figure presentation.
3. More in-depth discussion is needed to elaborate pros & cons of DSHI, SCELD and Voigt fitting.
4. There is a room for authors to improve the English presentation.
Round 2
Reviewer 1 Report
I am happy to approve this submission.
Reviewer 2 Report
The corrections were satisfactory. I agree to the publication of this paper.
Reviewer 3 Report
In the revised manuscript, I humbly believe that the authors have improved the manuscript to meet the minimum requirement to publish in Sensors